# Control of the Drying Patterns for Complex Colloidal Solutions and Their Applications

**DOI:** 10.3390/nano12152600

**Published:** 2022-07-28

**Authors:** Saebom Lee, Tiara A. M., Gyoujin Cho, Jinkee Lee

**Affiliations:** 1School of Mechanical Engineering, Sungkyunkwan University, Suwon 16419, Korea; leesb@skku.edu; 2Department of Biophysics, Sungkyunkwan University, Suwon 16419, Korea; tiara.am@gmail.com; 3Institute of Quantum Biophysics, Sungkyunkwan University, Suwon 16419, Korea; 4Research Engineering Center for R2R Printed Flexible Computer, Sungkyunkwan University, Suwon 16419, Korea; 5Department of Intelligent Precision Healthcare Convergence, Sungkyunkwan University, Suwon 16419, Korea

**Keywords:** coffee-ring effect, evaporation, interfacial flow, deposition patterns

## Abstract

The uneven deposition at the edges of an evaporating droplet, termed the coffee-ring effect, has been extensively studied during the past few decades to better understand the underlying cause, namely the flow dynamics, and the subsequent patterns formed after drying. The non-uniform evaporation rate across the colloidal droplet hampers the formation of a uniform and homogeneous film in printed electronics, rechargeable batteries, etc., and often causes device failures. This review aims to highlight the diverse range of techniques used to alleviate the coffee-ring effect, from classic methods such as adding chemical additives, applying external sources, and manipulating geometrical configurations to recently developed advancements, specifically using bubbles, humidity, confined systems, etc., which do not involve modification of surface, particle or liquid properties. Each of these methodologies mitigates the edge deposition via multi-body interactions, for example, particle–liquid, particle-particle, particle–solid interfaces and particle–flow interactions. The mechanisms behind each of these approaches help to find methods to inhibit the non-uniform film formation, and the corresponding applications have been discussed together with a critical comparison in detail. This review could pave the way for developing inks and processes to apply in functional coatings and printed electronic devices with improved efficiency and device yield.

## 1. Introduction

Deposition patterns from an evaporating droplet containing solute or colloidal particles leave a ring-like stain at the droplet periphery, termed the “coffee-ring effect”. Colloidal solutions, which consist of nano/microparticles dispersed in solvents, are useful for a variety of technological applications such as printing [1,2,3,4], coating [5,6,7,8,9,10,11,12,13,14,15], micropatterning [16,17,18] and bio arrays [19,20,21] owing to their functionalities. For example, quantum dots (QDs) are semiconductor nanoparticles with tunable optical properties depending on their size and photochemical stability and have been proven suitable for the light-emitting display industry [22,23]. Mixtures containing silver, graphene nanoparticles, carbon nanotubes (CNTs), etc. are used in the fabrication of electronic devices due to their high conductivity, compatibility with substrates and excellent mechanical properties [24,25,26,27,28]. During the manufacturing process, the printed film displays non-uniform 3D morphology after solvent evaporation due to the coffee-ring effect, leading to low-resolution patterns and challenges in reproducing similar patterns, subsequently decreasing the performance of the device. Therefore, it is important to understand the dynamics of dispersed particles in an evaporating sessile droplet, to control the desired morphology. Many studies have demonstrated that the final pattern is influenced by the evaporation mechanism [29,30,31,32,33,34,35,36], flow dynamics inside the droplet [37,38,39,40] and multi-body interactions [41,42,43,44,45,46]. It was observed that a variety of patterns can be formed by controlling these effects, showing spider-web [47,48], multi-ring [49,50,51] and spot-like [52,53] deposits as well as homogeneous morphology.

Researchers have theoretically and numerically developed the dynamics of an evaporating droplet for the past few decades and suggested methods to control the deposit pattern using chemical additives, external sources, variation of particle concentration or size, and different geometries. The mechanism of the ring formation from a drying sessile droplet was described when the contact line is pinned on the substrate, and theories were proposed to predict evaporation rate and flow velocity inside the drying droplet in the case of a pinned contact line [29,30]. Several studies explored the flow field within droplets as a function of time [38,39], the effect of Marangoni flow, and the structural transition of colloidal particles in the stain during droplet evaporation through numerical and experimental analysis [54]. Though the formation of coffee-ring is ubiquitous, several methods have been proposed to overcome the formation of the ring-like patterns. Chemical additives including polymers [55,56,57,58,59], surfactants [60,61,62,63,64] and salts [42,65,66,67,68,69,70,71,72] alter the rheology of the droplet, thereby changing the pattern formed after droplet drying. This method has been widely used due to simple and effective control of the deposition but remains limited in biochemical applications because it causes contamination. Other strategies without modifying the composition of the droplet have been introduced, where drying conditions [73,74,75,76,77,78,79,80,81,82] or geometry [83,84,85,86,87,88,89,90,91,92,93,94,95,96,97,98,99,100] can be changed. Recent works have reported the effect of external sources including external vapor [101,102,103,104,105,106], magnetic field [107,108], and surface acoustic wave (SAW) [52,109] on the deposit patterns.

In this paper, the authors describe the basic evaporation mechanism of a sessile droplet, and how Marangoni flow can modulate the coffee-ring formation and evaporation modes depending on surface wetting properties to better understand the methods to control final deposit patterns. Next, the paper reviews various techniques including classical and newly developed methods for the controlling pattern formation of drying droplets, namely chemical additives, external factors, and geometrical configurations, and finally explores relevant applications.

## 2. Evaporation of a Sessile Droplet

The ring-like stain observed after the drying of colloidal solutions is a common occurrence as shown in Figure 1a, which is known as the coffee-ring effect caused by the non-uniform evaporation rate along the droplet surface. Studies have investigated the evaporating sessile droplet to understand the mechanism of the ring formation at the edge and emphasized the effect of contact line pinning on it. The coffee-ring effect is modulated with Marangoni flow induced by recirculating flow in the droplet, and it is required to comprehend evaporation modes depending on surface wettability to understand various deposition patterns.

### 2.1. Evaporation Mechanism of a Sessile Droplet Pinned on a Substrate

The ring formation is caused by an outward flow in the droplet which is called capillary flow. The rate of evaporation is faster near the edge as compared to the center of the droplet (Figure 1b) so that the capillary flow is induced by replenishing liquid lost by evaporation during contact line pinning [29,31,37]. Researchers investigated the variation of contact angle with contact line pinning during evaporation of the droplet, and theoretically predicted the fluid velocity and the growth rate of the ring [29,37]. For an evaporating droplet pinned on a substrate, the relationship between vertically averaged radial flow, droplet height over time, and rate of mass loss due to evaporation are calculated from the mass conservation of fluid. The amount of fluid at a radial distance is equal to the sum of the net flux of liquid into the control volume and the amount of mass loss by evaporation from the surface of the element. Here, the droplet is assumed to have an axisymmetric spherical shape, and the gravitational force is not considered because the surface tension force is relatively greater. When the evaporation is induced by the liquid-vapor diffusion above the droplet surface, the evaporation rapidly approaches a steady-state such that the diffusion equation boils down to the Laplace equation.
(1)$$\frac{\partial c}{\partial t}=D\nabla^{2}c\approx 0$$
where c is defined as the mass concentration of local water vapor and D is the diffusion coefficient for vapor in the air. The boundary conditions are that (1) c is saturated vapor concentration $c_{s}$ along the droplet surface, (2) the vapor mass flux J normal to the substrate is zero at the substrate and (3) c is the ambient vapor concentration while $c_{\infty }$ is away from the droplet. Equation (1) is solved to evaluate the distribution of vapor above the droplet and evaporation flux. The local evaporation flux at the droplet surface is defined as $J_{s}\left(r,t\right)=-D\nabla c$ and the total evaporation rate along the droplet can be obtained by integrating $J_{s}\left(r,t\right)$ over the surface area of the droplet [31]. Vapor flux is calculated along the droplet surface during evaporation with a finite element method (FEM). Figure 1b depicts an increase in the flux from the top of the droplet center to the contact line at the edge. The flux is theoretically infinite at the edge as shown in the inset of Figure 1b. Finally, the study suggested an approximate expression using the FEM for the droplet evaporation rate as a function of the contact angle for 0° < *θ* < 90°. These studies give a better understanding of how to predict the droplet volume change, the fluid flow velocity within the droplet, the growth rate of the ring, and so on for an evaporating sessile droplet pinned on a substrate.

### 2.2. Marangoni Flow

The capillary flow in a radially outward direction leaves the ring-like deposits of included particles at the droplet edge. This edge deposition can be controlled by recirculating flows or inducing additional flows in the droplet [39,52,110]. Marangoni flow is driven by the surface tension gradient induced by mixing solvents or adding a surfactant, where liquid flows away from regions of low surface tension [111]. In general, the Marangoni flow acts in a direction opposite to that of capillary flow such that it creates a recirculatory flow inside the droplet and transports the solute from the contact line. Figure 1c represents a comparison of the deposits by capillary flow and Marangoni flow [112]. The capillary flow from the droplet center to the edge was seen in the case of water droplets containing a small number of bacteria and caused the contact line deposits. In contrast, a water droplet with high amounts of surfactants created a gradient in surface tension and showed vortices near the droplet edge. The outward flow carried the bacteria to the droplet edge, and the Marangoni flow pulled the bacteria upwards and inwards to the center of the droplet. Thus, these vortices inside the droplet resulted in the inhibition of the coffee-ring.

During the droplet evaporation, thermal Marangoni flow occurs because the surface tension varies along the surface of the droplet due to temperature differences by evaporative cooling. A study numerically and theoretically proved the flow recirculation induced by temperature gradient [113]. However, it was shown that the thermal Marangoni flow is not significant for the experiment with drying water droplets because the contaminants along the free surface can suppress the Marangoni flow. Marangoni flow is also driven using binary mixtures of the components with different surface tension and volatilities as shown in Figure 1d [114]. During the evaporation of a sessile droplet containing a binary mixture, the component with the higher volatility is depleted from the contact line region due to faster evaporation, following which the resulting surface tension gradient drives the solutal Marangoni flow. For the droplet pinned on a substrate, these flows can create homogenous deposit patterns.

### 2.3. Effect of Surface Wetting Characteristics on the Evaporation Process

Furthermore, surface wetting properties influence the deposition patterns of an evaporating droplet. For evaporating droplets on surfaces with different wettabilities such as hydrophilic and hydrophobic, there are two evaporation modes: constant contact radius (CCR) mode and constant contact angle (CCA) mode as shown in Figure 1e [115,116]. The CCR mode is mainly observed for a hydrophilic substrate. The droplet spreads out and is pinned on the substrate, after which the contact angle reaches the advancing contact angle value. During evaporation, the contact radius remains constant, and the contact angle decreases from the advancing contact angle to the receding contact angle value. In contrast, for a hydrophobic substrate, the CCA mode is dominant over the CCR mode due to low contact angle hysteresis (CAH) and weak contact line pinning. Here, the spreading process of the droplet is negligible. Thus, the contact angle remains constant, and the contact radius decreases with time. These different processes of evaporation will finally influence the deposition pattern of the droplet.

## 3. Chemical Additives

Amongst the different methods used to mitigate the coffee ring formation, the latest findings have emphasized how the external additives alter the flow dynamics and thereby neutralize the outward capillary flow [55,63,112,117,118]. The following sections describe the mechanism of polymer, surfactant, and salt addition along with pH modulation on the drying pattern of colloidal solutions in detail.

### 3.1. Effect of Polymer Additives

Researchers have been exploring the effect of polymer addition in altering the deposition pattern for different complex colloidal solutions [55,56]. Upon the addition of a polymer, it was noted that the solution viscosity is enhanced, thereby creating significant resistance to the radially outward flow. This, alongside the Marangoni effect, due to the variation in polymer concentration from droplet evaporation, results in a fewer number of particles to be deposited at the edge, which does not favor the pinning of the contact line [55,57]. The depinning of the contact line was facilitated due to the Marangoni flow upon the addition of polymer, as the force necessary for depinning seemed small [119].

The formation of a uniform film is crucial in pharmaceutical manufacturing technologies, particularly considering the distribution of active ingredients after drying [12,120]. One such example is shown in Figure 2a, which depicts the effect of chitosan polymer additive on the drying pattern of aqueous paracetamol droplets, specifically, the suppression of the coffee-ring effect [121], at different chitosan formulations. Upon drying on a glass substrate, the drug crystals recrystallize to develop an undesired coffee-ring pattern due to peripheral deposition. The paper reported that an increase in viscosity by chitosan addition inhibited the crystallization of the drug droplet by preventing the critical mass required to form the nuclei through solute immobilization [55]. It was observed that the crystalline drug deposits were formed only in the case of F0 formulation, where no chitosan polymer was present. When the concentration of chitosan polymer was increased, a gradual shift to a more spherical shape was also seen due to an increase in viscosity, leading to the non-spreading or beading of the droplet on the substrate [37,121]. Initially, the polymer distribution in the solution is uniform, with a constant volume fraction throughout. As the drying process commences, the polymer volume fraction starts to increase leading to an increase in dispersion viscosity. However, there exists an optimum value due to the onset of polymer gelation [122,123].

A numerical investigation of the evaporation of droplets revealed that polymers with higher molecular weight mitigated the coffee-ring effect, reducing the peak ring to the final central height ratio [56]. It was noted from the literature that the molecular weight of the polymer species with a polydispersity index greater than 1, had a strong contribution to the viscosity value [124,125]. In Figure 2b, optical microscope images of the magnified edge and droplet center after droplet evaporation are shown, with and without the addition of different polymer additives, namely polyethylene oxide (PEO1, of molecular weight = 4 × 10^6^ and PEO2, of molecular weight = 3 × 10^5^) and polyvinyl alcohol (PVA, of molecular weight = 3.1 × 10^4^ to 5 × 10^4^). In all the cases, the relative viscosity was observed to be greater than 1, the value being different for each additive due to the variation in polymer–SiO_2_ microsphere, polymer–substrate or polymer–chain structure interaction, during droplet evaporation [126,127].

### 3.2. Effect of Surfactants

Surfactants decrease the surface tension of the solvent (e.g., solvent-like water) [60,61]. As the liquid evaporates, the concentration of the surfactants increases locally, generating strong gradients in the surface tension [62]. The presence of surfactant additive in a solution is also noted to bring in a localized central deposition than at the edges, leading to depinning of the contact line [62]. The surfactant-induced Marangoni flows were found to generate a novel quasi-steady-state “eddy” that inhibits the majority of the particles from reaching the contact line, thereby causing the depinning of the contact line as evaporation proceeds [63].

Furthermore, the nature of surfactant (ionic or non-ionic), or the electrostatic interactions amongst surfactants and particles is considered important in determining the residue shape [43]. Adding cationic/anionic or non-ionic surfactants have different effects on the final deposition/morphology patterns. The addition of non-ionic surfactant maintains the self-assembly of colloidal spheres similar to that of an additive-free solution, with the charged particle/substrate system having an ordered structure at the edge of the substrate and oppositely charged particles/substrate having a disordered arrangement [129]. Upon the addition of an ionic surfactant (cationic/anionic), the extent of the particle–substrate Coulombic interaction could be modified by lowering the surface charge density [130].

Figure 2c illustrates the effect of cationic (hexadecyltrimethylammonium bromide-CTAB, dodecyl trimethylammonium bromide-DTAB) and anionic (sodium dodecyl sulfate-SDS) surfactants on the final deposition pattern of a droplet containing anionic particles. For a similarly charged particle–surfactant system as in the case of SDS surfactant, the drying proceeds to form a coffee-ring pattern for all the concentrations taken, for concentrations less than the critical micelle concentration. The reason behind this is the electrostatic repulsion between the negatively charged polar head of SDS and the anionic polystyrene (PS) particles [43]. In the case of dissimilar charged systems, the adsorption of surfactant affects particle–liquid/gas interface interactions, thereby controlling the deposition pattern. At low additive concentrations, ring-shaped deposits were formed after drying. An intermediate surfactant concentration promotes particle trapping at the interface, leading to a homogeneous pattern. However, for high additive concentration, the coffee-ring effect is observed due to charge reversal [131].

Researchers have studied the effect of adding amphiphiles on the deposit patterns formed by silica nanoparticles by altering the wettability and stability [128]. Figure 2d shows the different drying patterns of a droplet, due to the adsorption of short-chain amines by the silica nanoparticles. For high wettability with small zeta potential values, the droplet spreading led to a branched structure formation. A medium wettability and negative values of zeta potential led to edge deposition resulting in the formation of a ring-shaped deposit. For a more hydrophobic condition and small zeta potential value, the particles formed a dot-shaped deposit through sedimentation. The variation in the deposition pattern with the alkyl chain length can be accounted for by considering the effect of amine adsorption on the particle–particle interactions and the particle–substrate interactions [117,128].

### 3.3. Effect of Salt Addition

From the literature, the significance of electrostatic forces in influencing patterned colloidal depositions has been evident [132]. The variation in the DLVO (Derjaguin–Landau–Verwey–Overbeek) interactions between particles, and particle–substrate affects the shape of the deposits, with an attractive force transporting particles towards the substrate, resulting in a uniform deposit formation [42].

The addition of salt alters the interactive forces due to electrostatic screening effects which in turn alter the shapes of the deposits [66]. To achieve a uniform deposition pattern and scalable control of the coffee-ring effect, researchers have added trace amounts of salt, namely chloride salts of Na, Li, K, Ca, or Mg to the sample solution [67]. Figure 3a shows the schematic representation of how the presence of cations determines the deposition of aqueous droplets containing PS microspheres on graphene (i), and the pattern after drying on graphene as the salt concentration increases (ii–v). The uniform pattern formed may be attributed to the strong cation–π interactions amongst the hydrated cations and the aromatic rings on the substrate surface, which is not thereafter disturbed by the flow pattern to the droplet boundary [68,69,70].

Figure 3b shows a comparison of the thickness at the edge and center of the PS microsphere sessile droplets upon evaporation, before and after the addition of Zn^2+^. The cations are dissolved to form a complex with ethylenediamine, as the theoretical calculations showed that suspended materials were uniformly adsorbed on the surface-mediated by complexed cations through strong cation–metal and cation−π interactions [71]. As the concentration of the complexed Zn^2+^ increases up to ~12.0 mM, the center and edge thickness almost become equal, due to the cation control of the coffee-ring effect. Studies have also revealed that the presence of salt affects the evaporative self-assembly process by regulating the strength of particle–surface electrostatic interactions owing to the formation of electric double layers [72]. Researchers have come up with the idea that the ring-like deposition can be counter-acted via an opposing electric field by adding specific salts at millimolar concentrations, which causes diffusiophoretic and diffusion-osmotic flows in the system [133]. An increase in ionic concentration as evaporation proceeds rapidly generates a radial salt concentration gradient which produces a spontaneous electric field within the droplet causing electrokinetic transport of the particles and fluid near the substrate surface, permitting the control of deposition pattern.

### 3.4. Effect of Variation in pH

From the literature, several notable studies were reported that dealt with controlling the pH of the liquids to overcome the coffee-ring effect [117,118,134]. It was revealed solution pH affects the deposition pattern due to the force interactions between substrate and particles, namely the electrostatic and Van der Waals forces [117]. Figure 3c depicts the effect of the pH-dependent DLVO interactions between the particles (titania) and the substrate (glass) on the deposition morphology. It was observed that a ring-shaped deposit is formed initially due to contact line pinning, which changes into a uniform pattern upon changing to an acidic condition (pH = 1.4) due to the attractive DLVO forces between the particles and substrate. At intermediate pH, the pattern consists of aggregates covering the entire initially wetted area, due to the weak particle–substrate DLVO force as compared to the interparticle Van der Waals attraction. As pH increases, the strong repulsive DLVO force amongst the substrate and the particles pushes the particles away from the interface along with the main capillary radial flow to the droplet contact line [117].

Figure 3d shows the pattern formation for different particles under acidic and basic conditions. Carboxyl-PS particles display a clear ring-like pattern to an almost uniform monolayer deposition when the pH value is altered. On the other hand, sulfate-PS forms a uniform monolayer pattern under acidic and basic conditions [118]. The presence/absence of H^+^ and suppression/enhancement of R-COO^-^ in the carboxyl group at acidic and basic conditions lowered and increased the zeta potential, respectively, thereby supporting the hypothesis. In the case of sulfate-PS particles, the presence of fewer functional groups makes the particles less hydrophilic, thereby self-assembling them into a monolayer pattern at the liquid-solid interface. In contrast, the presence of more functional groups on the particle surface makes carboxyl-PS particles more hydrophilic, developing self-assembly of particles for zeta potential of −7.2 mV or lower, equivalent to a pH of 3 or less.

## 4. External Factors without Chemical Additives

Several studies have also demonstrated various methods to control the final deposition pattern of an evaporating droplet without modification of its composition. It can be achieved by altering the dynamics of internal flow using external factors such as vapor source [101,102,103,104,105,106], atmospheric drying condition [73,74,75,76], the temperature of substrate [77,78,79,80,81,82], SAW or magnetic field [52,135,136]. These factors help to obtain uniform or spot-like dried patterns of colloidal droplets. This section describes the deposit patterns controlled by strengthening the internal flow inside the droplet with external sources in the absence of chemical additives.

### 4.1. External Vapor Source

Recent works have reported dried colloidal deposits by simply changing local surface tension on the droplet’s upper surface using an external source of vapor. In general, if an evaporating sessile droplet and the pendant droplet which is used as a vapor source have the same component, the evaporation of the droplets is suppressed through vapor mediation interaction with each other [101]. However, in the case of droplets with different components, the pendant droplet with higher volatility evaporates and adsorbs on the surface of the other, driving a Marangoni flow due to the local surface tension gradient. As shown in Figure 4a(i), a needle containing ethanol acts as a vapor source near the surface of a water droplet during evaporation and it results in a difference in surface tension between the edge ($\gamma_{1})$ and top ($\gamma_{2})\;$ of the droplet where $\gamma_{1}>\gamma_{2}$ [102]. As a result, recirculating Marangoni eddies in the droplet suppress the coffee-ring effect on the final deposit pattern. Furthermore, this study explored spatially controlled Marangoni flow by lateral displacement of the vapor point source and analyzed corresponding vorticities and dried patterns (Figure 4a(ii,iii)). When the vapor source moves from the center toward the edge of the droplet, flows are strengthened near the edge. Here, the stagnation point of the flow is also displaced toward the edge and thus asymmetric flows in the confined geometry of the droplet generate vorticity below the vapor source. Correspondingly, particles are accumulated at the center forming spot-like deposits or near one side of the edge with an asymmetric ring.

Another study investigated the splitting of an elongated sessile droplet using vapor-mediated interaction, which is called the “Moses effect” [103]. As shown in Figure 4b(i), vapor from the pendant droplet of ethanol is adsorbed on the surface of the long water droplet and Marangoni flow splits the central section of the droplet into two halves. Figure 4b(ii) shows the comparison of particle deposits with and without the Moses effect. In the absence of an ethanol vapor source above the droplet, graphite nanoparticles having low density initially float on the surface of the droplet and then move toward the substrate during shrinkage of the droplet only due to evaporation. Thus, the coffee-ring effect is not observed, forming a uniform deposit pattern. However, when evaporation and Marangoni effect work together due to the vapor source surrounding the droplet, the interface in the center regresses downward and particles are drained from the central section. Here, the rate of particle settling determines the deposit pattern, tripartite of a three-part deposit or bipartite pattern of a two-part pattern. In addition, Marangoni flow by vapor-mediated interaction can be utilized to disperse the central aggregates of buoyant hollow spheres [104].

### 4.2. Relative Humidity

The drying environment influences the deposit pattern in the absence of chemical additives because it modulates the dynamics of the internal flow in a drying droplet. It can be achieved by varying relative humidity (RH) of drying conditions, and a few studies showed the control of the dried patterns under various RH levels [73,74,75,76]. Figure 5a represents the deposition mechanism of SiO_2_ particles in a sessile water droplet in a range of 33% to 75% RH [73]. For a lower RH of 33% and 44%, the conventional ring-like deposits are formed but as the RH increases up to 75%, the microsphere distribution is more uniform. These results come from different internal flows in the evaporation environment. In an evaporating droplet, capillary flow occurs because of the variation in the evaporation rate near the center and droplet edge, and a very weak inward flow which is consistent with Marangoni flow is also induced on the surface of the droplet due to the temperature gradient, as described in Section 2.2. The Marangoni flow is strong at the beginning of evaporation, but rapidly diminishes, and the capillary flow becomes dominant over time. As a result, at a lower level of RH, only the capillary flow influences the final deposit showing the ring-like form. In contrast, the capillary flow in the drying sessile droplet is suppressed under a higher level of 63% RH and it carries fewer microparticles toward the edge of the droplet. Moreover, studies have dealt with the formation of a dried pattern of biological fluids depending on the RH [74,75]. The RH of the evaporating environment determines the spreading behavior and evolution of crack patterns of whole blood droplets [75]. After the complete evaporation of the blood droplet, the pattern is characterized by a central part with a sticking deposit and small cracks, and a fine periphery adhering to the substrate. The adhering region and crack nucleation strongly depend on the RH levels due to competition between the evaporation rate and the adhesion of a gel on the substrate.

### 4.3. Temperature

The direction of thermal Marangoni flow was examined in evaporating sessile droplets depending on the ratio of substrate conductivity $k_{s}$ and liquid conductivity $k_{L}$ [77]. If $k_{s}/k_{L}$ > 2, the Marangoni flow is induced radially outward along the substrate. The flow direction depends on the contact angle for $1.45<k_{s}/k_{L}<2$ and the flow directs radially inward for $k_{s}/k_{L}<1.45$. It was also observed how heating the substrate modifies the final deposit pattern of droplets. A thinner ring and larger volume of the central stain are observed with increasing temperature of the substrate due to a more complex flow of an evaporating sessile droplet, as represented in Figure 5b [78]. The mechanism of dried deposit formation was compared on uniform temperature substrate and non-uniformly heated substrate using particles of various sizes of d= 0.1, 1 and 3 μm [79]. On the substrate at a temperature of 60 ℃, the temperature of the surface near the droplet center is lower compared to the contact line owing to a thermal conduction resistance across the droplet thickness. It leads to a surface tension gradient along the liquid–gas interface and consequently, an axisymmetric thermal Marangoni flow occurs toward the apex of the droplet from the contact line as shown in Figure 5c(i). A stagnation region near the contact line is created forming a ring due to Marangoni flow, but most of the particles move inward. Furthermore, the final deposit pattern depends on the particle size. For smaller particles with a diameter of 0.1 μm, non-axisymmetric inner deposits are formed because of contact line depinning with a stick-slip motion, while larger particles with diameters of 1 and 3 μm develop ring-shaped axisymmetric deposits. A non-uniformly heated substrate generates a temperature gradient on the surface of the droplet; thus, Marangoni flow is induced from a higher temperature region $T_{H}$ to lower the temperature region $T_{L}$, as shown in Figure 5c(ii). In addition, the difference in evaporation mass flux due to varying temperature makes stronger Marangoni recirculation inside the droplet from $T_{L}$ to $T_{H}$. The result of the final deposits shows a thicker ring on $T_{L}$ side for smaller particles of 0.1 μm in diameter but a thicker ring on the $T_{H}$ side in the stagnation region for larger particles with diameters of 1 and 3 μm.

### 4.4. Other Factors

Besides these external sources such as vapor, humidity and temperature, other factors also affect the final patterns of drying droplets in the absence of chemical additives. Recent work suggested new insight into how the addition of air bubbles in a colloidal droplet controls the deposition patterns [135]. Figure 6a (i,ii) represents the entire drying process of a droplet on a monocrystalline silicon wafer, without and with air bubbles respectively. The drying process of the bubble-free droplet is divided into three stages where the ring-like deposit grows, and the contact angle droplets at the first stage (0–299 s). In the second stage (299–357 s), the contact line retreats, and a ring-like pattern is finally formed during the third stage at 357–367 s. However, the drying sequence of the droplet containing bubbles can be divided into two stages. Initially, the bubbles float at the top center of the droplet by buoyancy force and its contact angle reduces during evaporation (0–213 s). Then, in stage II (213–350 s), the contact line retreats smoothly, forming a uniform deposition pattern. These phenomena are related to contact angle behavior. The bubble-free droplet forms a convex shape, and its initial apparent and receding contact angles are ~17° and ~8°, respectively. In contrast, the droplet with bubbles shows initially a concave-shaped meniscus due to the buoyant bubbles (Figure 6a(iii)), and the initial apparent contact angle is smaller, but the receding contact angle is larger compared to those of the droplet without bubbles, which are measured as ~14° and ~11°, respectively. Thus, these contact angle behaviors cause fast and smooth retraction of the contact line and finally leave more uniform patterns.

Another strategy to achieve uniform deposits is utilizing a light source to actuate Marangoni flow in the droplet [136]. The light source rapidly converts light energy into heat and induces a nonuniform temperature distribution along the droplet surface. Figure 6b(i) shows the light-induced droplet evaporative crystallization of calcium sulfate (CaSO_4_) on hydrophobic and hydrophilic surfaces. The laser beam was irradiated to the droplet center, and the photothermal effect generates the internal flow in the droplet with increasing temperature, where the temperature is high near the focused light region. CaSO_4_ starts to crystallize at the interface near the focused laser region during evaporation and the formed crystals are not accumulated at the droplet edge by the effect of Marangoni flow. Thus, the final deposit forms a condensed pattern with a continuously decreasing interface of the droplet on the hydrophobic PDMS surface. Natural and plate-heated CaSO_4_ evaporations cause the ring formation because initially, the evaporation is in the CCA mode due to low contact angle hysteresis, and later changes to the CCR mode forming the crystals near the contact line. The light-induced condensed crystallization is also achieved on the hydrophilic surface. As shown in Figure 6b(ii), the deposits using the light-caused mechanism were compared to those induced by natural evaporation and uniform plate heating. Since the droplet is almost flat on a hydrophilic surface unlike on a hydrophobic surface, the crystals are first formed and deposited in the laser beam region after the evaporation.

In addition, a study analyzed the influence of SAW on the inhibition of ring-like deposit formation [52]. Applying the SAW causes particles to be confined at the nodes generated by standing acoustic pressure waves and capillary waves in the droplet, as described in Figure 6c. As time proceeds, the particles are formed along the nodal circles of the capillary waves, which are the thick concentric accumulations of the particles near the interface. Here, they confirmed the scaling relation of log (λ) vs. log (sinθ), with λ and θ being the wavelength and droplet contact angle, respectively, showing that the nodal circle is dependent on the capillary waves. Sweeping the SAW frequency changes the nodal point position by the intersection of the standing acoustic pressure waves and standing capillary waves. As a result, spot-like deposits or uniform disc-like patterns can be obtained by controlling the SAW frequency.

Moreover, under an applied static magnetic field, the deposit patterns of evaporating droplets have been studied. Figure 6d shows the completely dried patterns of ferrofluid droplets near the edge for the field strengths from 0 to 0.15 T when the field is applied parallel to the plane of the substrate [107]. Lower field strengths of 0.005 T and 0.02 T result in a thick ring pattern and chains at the droplet edge. As the field strength increases, the particles do not form chains and are concentrated at the droplet center. The results for the applied field perpendicular to the substrate are similar, showing decreased ring thickness with increasing field strength.

## 5. Effect of Geometry

Along with the influence of external factors, the deposition patterns of colloidal droplets at different configurations and geometries are also significant in understanding particle deposition physics [83,84,85,86]. As the total time taken for drying is dependent on the total surface area available for evaporation along with the evaporative heat flux, which is in turn dependent on the droplet shape, the droplet configuration plays a major role in deciding the deposition pattern [87]. The presence of ring-shaped patterns has been reported after the evaporation of colloidal droplets at different geometrical configurations, namely sessile droplets on a single substrate [88,89,90,117], and droplet evaporation on two substrate configurations [137,138,139] and hanging droplet configurations [140]. When the droplet configuration changes from the sessile arrangement, the force of gravity also comes into the picture, affecting the final deposition pattern [141]. When the droplet evaporates in a confined geometry, evaporating vapors are stagnated near the droplet interface and it eventually changes evaporation dynamics due to varying concentration differences, near and away from the interface [91,92,93,94,95,96,142]. In addition, substrate properties such as roughness influence the ring formation with cracks appearing at the periphery of droplets [99]. A few studies have compared the deposit patterns on the smooth and rough surfaces and analyzed the film morphology obtained after evaporation, considering the substrate roughness and other effects [100]. This section describes the role of geometry on the deposition patterns of colloidal droplets on an inclined substrate, pendant droplets, and sessile droplets in a confined system and on a rough substrate.

### 5.1. Inclined Substrate

Although the deposition pattern of colloidal droplets in the sessile configuration has been widely studied, in a majority of real-world scenarios, the drying of droplets occurs on substrates inclined at an angle to gravity [143]. In such configurations, the force of gravity on the particles, along with the gravity-aided droplet deformation affects the kinetics of evaporation, particle motion and thereby the final pattern morphology [87]. The force of gravity also alters the pinning-depinning mechanism of the contact line, thereby affecting the evaporation dynamics [144]. The droplet placed on a flat substrate tilted at an inclined angle is deformed by gravitational force, and the initial contact angle is divided into front and rear contact angles through inclination [145]. Due to a rapid change in the instantaneous contact angle, the upper contact line experiences a higher depinning force, and consequently, the upper contact line recedes rapidly as compared to the lower contact line. Thus, the evaporative flux which decreases with an increase in contact angle has a higher value close to the upper contact line in comparison to the lower contact line. This results in a higher solvent depletion from the upper contact line, altering the depinning dynamics on both sides of the droplet [87]. If the altered depinning dynamics cause a considerable reduction in the available evaporation area, then the time taken for the droplet evaporation on the inclined surface is higher, leading to an increased droplet lifetime on inclined surfaces [145].

A numerical analysis of the deposit variation due to the substrate inclination revealed that solute deposition depends on the initial volume of the droplet and the substrate inclination [146]. In the case of larger droplet volume and steep inclination, the deposit formed initially at the upper contact line is depleted because of the early depinning of the upper contact line and subsequent continuous deposit growth at the lower contact line. Alternatively, at low droplet volume and slight inclination, the solute deposition continues to grow at the upper contact line, as the growth rate is higher than downslope deposition [146].

Figure 7a(i) shows the schematic of the experimental setup used to analyze the kinetics of droplet evaporation and resulting deposition patterns at different inclinations of the substrate (inclination angle Φ = 0°, 45°, 90°, 135° and 180°), during the drying of an aqueous dispersion of PS particles. The authors found that as the angle of inclination increases from 45° to 135°, the deposit height and width at the advancing side of the droplet increase in comparison to the receding side, inferring that deposition asymmetry is a non-monotonic function of inclination angle [147]. Figure 7a(ii) analyzes the role of particle velocity components—parallel and perpendicular—in the advancing ($v_{∥}^{adv}$ and $v_{⊥}^{adv}$) and receding sides ($v_{∥}^{rec}$ and $v_{⊥}^{rec}$), respectively, on the particle spatial distribution in the deposits from droplet drying. For an inclination angle (Φ) of 0° or 180°, i.e., sessile, or pendant configuration, the particles stay in the corresponding halves of the droplet during the evaporation process, except for diffusive transport. However, in the case of any other orientation, a small particle fraction moves from the receding side to the advancing side, subsequently to the contact line by the advective velocity component. The parallel velocity components of the particles on the advancing and receding sides are given by Equations (2) and (3), respectively.
(2)$$v_{∥}^{adv}=v_{A\;}+v_{g}\sin\Phi$$
(3)$$v_{∥}^{rec}=v_{A\;}-v_{g}\sin\Phi$$
where $v_{A\;}$ and $v_{g}$ are the advective velocity which pulls the particles towards the contact line, and the gravitational velocity, with which the particles migrate vertically downward, respectively. From Equations (2) and (3), it can be concluded that for an inclination of 0° < Φ < 180°, the velocity of the particle parallel to the surface on the advancing side (lower half) is larger than the receding side (upper half) of the droplet, meaning that more particles arrive at the contact line on the advancing side at any given instant. Similarly, the components of particle velocity normal to the substrate are given by Equations (4) and (5), respectively.
(4)$$v_{⊥}^{adv}=-v_{g}\cos\Phi$$
(5)$$v_{⊥}^{rec}=-v_{g}\cos\Phi$$

From Equations (4) and (5), it can be concluded that the normal velocity component of the particle remains constant irrespective of where the particles are positioned. However, due to the change in sign of cosine function from 0° to 180°, the normal components take the particles towards the substrate below 90°, and away from the substrate above 90°. Consequently, more particles can be assumed to arrive at the advancing side of the droplet at Φ = 135° than at Φ = 45° [147].

### 5.2. Pendant Drop

The force of gravity played a more important role in particle transport in pendant droplets, as compared to surface tension or substrate roughness [148]. The deposition morphology was observed to be a result of the balance between gravitational sedimentation, shrinkage of liquid–gas interface and capillary flow directed radially outwards, especially for pendant droplets [149]. For a pendant droplet, the dried deposit pattern varies from coffee stains to central bump-like deposits with an increase in particle diameter and decrease in wettability [84,149,150], owing to gravity-assisted sedimentation and curvature-driven migration, i.e., the motion of particles that reach the contact line along the interface due to gravity, thereby suppressing coffee-ring formation.

Figure 7b(i) shows the sessile and pendant droplets with an initial contact diameter of 8 mm on a glass substrate and the dried patterns at different concentrations of 0.0001 wt.%, 0.01 wt.% and 1 wt.% for a Fe_2_O_3_/water suspension. The 22 nm nanoparticles were found to form rings at low concentrations whereas they formed spot-like patterns at higher concentrations in the pendant mode [141,151]. In contrast to the solute deposition at/near the ring observed in the case of a sessile droplet, for a pendant droplet, most of the solute particles were deposited in the interior of the ring. This difference was attributed to the particle aggregation dynamics in the suspension during the evaporation process, gravitational force, and variation in the geometrical constraints of the particle motion. In Figure 7b(ii), the ratio between the area of deposition and the hydrophilic area, *S*/*S*_0_, for different values of $Pe_{G\;}$ or gravitational Peclet number is depicted for a sessile droplet (indicated by blue line) and pendant droplet (red line) [149]. $Pe_{G\;}$ is defined as a function of particle size and gravitational acceleration, as per Equation (6) and can be used to analyze the relative effect of gravity on particle sedimentation as compared to thermal or Brownian motion [84].
(6)$$Pe_{G\;}=\frac{\pi d^{4}g\Delta \rho }{12k_{B}T}$$
where d represents particle diameter, Δρ, the difference in density between particles and the surrounding fluid, g is the acceleration due to gravity,$\;k_{B}\;$ denotes the Boltzmann constant, and T represents the temperature. With an increase in $Pe_{G}$, *S*/*S*_0_ was almost constant for a sessile droplet, whereas it was reduced in the case of a pendant droplet. For $Pe_{G}$ values > 10^−2^, sessile and pendant configurations have different deposition morphologies, with a ring-like stain observed in the case of the former, and an inner deposition pattern in the case of the latter, which became more concentrated as $Pe_{G\;}$increased. In the sessile configuration, the colloidal particles settled down near the substrate, hence particle deposition was mainly on the substrate, forming a comparatively uniform pattern. However, in the pendant configuration, the microparticles moved to the liquid–air interface and were trapped in the center. Hence, those trapped particles were transported from the liquid–air interface to the substrate and formed inner deposits [149].

Figure 7b(iii) depicts the final deposition patterns following the evaporation of colloidal hematite ellipsoids on four distinct substrates in the sessile and pendant configurations [143]. For a contact angle ≤ 90°, the sessile droplet leaves a coffee-ring pattern after drying, whereas suppression of the edge deposition is noted on a moderately hydrophobic substrate (θ = 120°). For a pendant droplet on a highly hydrophilic substrate (θ = 12°), the coffee-ring pattern is visible; however, as the contact angle increases, the deposition at the center becomes higher in comparison to the droplet edge, termed the coffee-eye [140,151]. The cause of the central deposition was attributed to particle aggregation and consequent gravity settling [140] or interface-mediated particle transport, i.e., particle adsorption on the surface of the droplet during evaporation and the consequent particle movement along the interface to higher curvature regions [143]. In the sessile configuration, the capillary flow aided bulk particle transport and interface transport owing to curvature differences have a similar contribution and act in the same direction, adding up to form the coffee-ring. On the other hand, in pendant configuration, the advection due to bulk flow is pointed towards the contact line, whereas the interface-mediated transport points towards the apex of the droplet. Moreover, the deviatoric curvature (the variation between the two principal radii of curvature) increases from the contact line toward the apex, causing the particles to move away from the pinned contact line, leading to a dome-shaped deposit after drying [152].

### 5.3. Confined Geometry

Confinement alters the evaporation dynamics of a sessile droplet leading to the change in flow fields, evaporation timescale and evaporation modes in a droplet by vapors accumulated in the confined geometry [91,92,93]. Evaporation is typically driven by a concentration gradient between droplets and the atmosphere. In the confined system, evaporating vapors are trapped in the vicinity of the droplet, and it decreases the concentration difference, thereby resulting in increasing total evaporation time [92]. For the mixture of droplets with different volatility and surface tension, stagnant vapors not only change the evaporation flux but also flow patterns due to varying local surface tension along the droplet surface with time. A study observed how the confined system affects the internal flow in a drying ethanol–water mixture droplet and corresponding dried patterns [94]. Here, the increased ethanol concentration near the surface of the drying droplet alters the evaporation flux. When they compared the internal flows of the binary-mixture droplet in an open and a confined geometry as can be seen in Figure 8a(i,ii), complicated Marangoni flows were initially observed everywhere in the droplet and the inward circulation lasts for a longer time in the confined system. It is because the stagnant vapors near the interface induce a surface tension gradient that gradually diminishes. Correspondingly, relatively more particles are accumulated at the center of the droplet as compared to the open geometry. The deposit patterns of the droplet containing quantum dots are shown in Figure 8a(iii,iv). While the droplet evaporation in the open geometry causes the coffee-ring effect, the confinement leads to uniform patterns on both glass and silicon wafers. The effect of confinement has not been considered much, but some studies have recently investigated the evaporation dynamics or the deposition in different confinement geometries [95,96]. Colloidal droplet confined between parallel flat plates shows significant spiral patterns on both top and bottom substrates after the complete evaporation of the solvent. It results from the continuous stick-slip motion of the contact line and the effect of gravity is not shown here. The resulting deposit depends on the confinement spacing, volume and concentration of the droplet [96].

### 5.4. Substrate Roughness

Substrate properties are an alternative method to change the droplet evaporation mechanism. It was observed that the coffee-ring formation can be controlled with pillars on the substrate, pore size and roughness of substrate [97,98,99,100]. The rough surface especially plays a critical role in the crack formation in the deposition film of drying droplets. As shown in Figure 8b(i–vi), the crack formations of polytetrafluoroethylene (PTFE) colloidal droplets were observed on the substrates with different roughness, which was achieved with the sandpapers of different grit sizes [99]. On a smooth glass substrate, a wide and thick ring is formed near the edge, and radial cracks are distributed on the ring-shaped deposit (Figure 8b(i)). As the roughness increases, the width of the ring and crack quantity decreases, while the distance between the cracks increases as the increasing roughness enhances the coffee-ring effect. The rough surface shows a different evaporating mechanism of colloidal droplet compared to that on the smooth surface, and it is represented in Figure 8b(vii). The evaporation rate increases with a decreasing contact angle of the droplet pinned on the rough surface, which strengthens the capillary flow and causes more particles to move to the contact line. Therefore, the cross-section is changed from a wedge- to a hill-like shape, showing the decreasing width and increasing thickness of the ring near the edge and only a few particles at the center of the droplet. Furthermore, on the rougher substrate, the crack expansion rate is enhanced because the strengthened coffee-ring effect stores more elastic energy.

Another study reported the effect of the roughness coupled with particle size and concentration on the dried patterns, as shown in Figure 8c [100]. On the rough silicon surface obtained by chemical etching, a remarkable ring pattern is created due to the dominant CCR mode, and the stains have different morphology depending on the particle size and concentration. Cracks generally appear at the periphery due to evaporation-induced stresses. However, the deposit pattern of the drying droplets with higher particle concentration exhibits cracks even in the interior region and the density of the crack in the interior region increases as the particle concentration increases and the particle size decreases. Irregular cracks are formed at lower particle sizes (7 and 12 nm) and concentrations (0.007 and 0.3 vol%), while the crack morphology becomes more regular at a higher particle size of 22 nm or higher concentration with lower size. It results from the fact that the large size particles contain a few crack boundaries due to fewer defects and Brownian force being inversely proportional to the particle size. In addition, the drying droplets containing higher concentration shows delamination for small size particles.

## 6. Summary

In this paper, the recent advances in regulating the drying pattern of colloidal solutions, specifically the coffee-ring effect, have been reviewed. Based on the suppression of the coffee-ring effect, various deposition morphologies of colloidal droplets are engineered. To modulate the deposition behavior of drying drops, techniques involving the addition of chemical additives, application of external factors, and variation in geometrical configurations, specifically inclined substrates, and evaporation in pendant and sessile modes, have been employed, as described in the prior sections. Chemical additives regulate the deposition patterns by altering the solvent properties such as surface tension and viscosity and varying the type of DLVO interactions. For applications requiring no change in the fluid composition, internally recirculating flow can be induced by a surface tension gradient due to variation in the drying conditions, or controlled particle migration using external factors. Different geometrical configurations also allow the regulation of the deposition patterns because of gravity or by varying the evaporation rate and mode. Most of these methods require modifying the properties of substrate, particles or the nature of the liquid. To overcome the limitations in applications, novel strategies to control the coffee-ring effect have recently been developed using ice drying, bubbles, and so on, but the studies on manipulating deposit patterns without any modification of the properties are still progressing. Furthermore, for a wide range of advanced technologies, the methods to control particle migration in a large area coating [153] and multi droplets or patterns [154] would be needed.

The rapid progress of patterning methods and the need for specific morphologies are an indication of the growing demand for functional patterns and homogeneous assembly in various applications. In recent times, studies on taming the coffee ring for nanopatterning have emerged as an advanced research hotspot in the field of future materials and devices. Having a nanoparticle assembly that can be regulated during the evaporation process of colloidal droplets offers a method for manufacturing functional nanomaterial patterns on a substantial scale. This will be of great significance for real-life applications namely electronic skins, wearable devices, and stretchable conductors. The exceptional performances of nanomaterial patterns could pave a way for promising applications in functional device fabrication and 3D printing technology.

## 7. Applications

The final morphology of a colloidal solution after drying is important in evaluating the characteristics of the solution. For example, distinct patterns of the dried droplet of blood serum are formed depending on the individual’s health conditions, as described in Figure 9a [155,156]. Figure 9a(i,iii) represents the dried blood droplet pattern in healthy individuals, whereas Figure 9a(ii,iv) shows patterns of individuals who are suffering from a disease. The formation of high-quality micro-patterns through solvent evaporation is noted to enhance the performance of optical/electric devices [157]. How the solute deposition morphology pans out during drying is imperative in understanding and monitoring the deposition processes in several manufacturing applications of printing or coating process [12,13,14,153], fabrication of functional nanomaterials [158,159], colloidal crystals [160] and ordered structures [161].

The ubiquitous nature of the coffee-ring effect has made these processes difficult, resulting in heterogeneous functional patterns [67]. The formation of a heterogeneous pattern or ring-like deposit has also implications for the performances of applications as diverse as biochemical analysis [162], fractures through self-assembly [163,164], and combined movement of bacterial colonies [112,165]. Additionally, the presence of coffee-ring impedes the functions or sensitivities for applications such as fluorescent microarrays as shown in Figure 9b where a non-ideal axisymmetric distribution of cDNA is observed [166], matrix-assisted laser desorption ionization (MALDI) spectrometry [167,168], and surface-enhanced Raman spectroscopy (SERS) [169].

Numerous industrial applications use the control techniques discussed in the previous sections, to form a homogeneous pattern after drying of colloidal solutions. To obtain a uniform deposition in the case of a binary mixture, researchers used a small concentration of surface-adsorbed polymer along with a surfactant, offering a new physicochemical avenue for the control of coatings [119]. It was also observed that the ring-shaped structure commonly seen in protein microarrays could be eliminated by the addition of non-ionic surfactants [170]. The surfactants displaced proteins that were otherwise adsorbed at the liquid–gas interface and subsequently transported to the edge of the droplet. In certain functional electronic devices, the performance is enhanced by controlling the particle assembly by the addition of surfactants, without impeding the film quality and function [157]. Moreover, the specific additive concentration required for coffee-ring effect inhibition did not significantly affect the droplet’s geometry or the total evaporation time, making it a feasible method for industrial and laboratory applications, such as the print and paint industry and biological applications [171].

The presence of trace amounts of salts, especially cations further enhanced the color fastness of dyes, ensuring inexpensive and scalable production, making them suitable for use in the textile industry, packaging, displays and military applications, as shown in Figure 9c.

A homogeneous assembly of colloidal particles is formed by aligning the magnetic moment of the nanoparticles along the direction of the applied magnetic field, which finds application in photonics for structural color display [172,173] and coating the pseudo-spin valve (PSV) thin films in spintronics devices [3,174]. As the exposed area can be altered in the presence of a magnetic field, magnetic pattern printing also can be used in decorative or security printing [173]. The self-assembly of solute particles can also be regulated by interaction with SAW to make uniform disc-like or concentrated spot-like deposition residues, for use in diagnostics, printing, and mass spectrometry [52,175].

Utilizing an external vapor source for drying colloidal solutions is suitable for homogeneous deposition of catalyst nanoparticles to grow single-walled carbon nanotubes, along with the fabrication of plasmonic films of gold nanoparticles. The morphology can also be monitored by controlling the exposure time of polymer dots to the vapor atmosphere by switching between evaporation and condensation processes, which could be used for regulating the dot profile in inkjet printing technologies [176]. Likewise, temperature control of substrate was implemented to control the droplet’s behavior, particularly the coffee-ring effect by varying the heat flux at the droplet edge during inkjet printing [177,178,179].

The effect of the substrate configuration on the deposition morphology is an important aspect for various applications, namely the distribution of fertilizer drops and micronutrient spray deposition patterns on the leaves at different orientations, contamination adhesion to the exterior of buildings from pollutant-laden drops and forensic analysis to reconstruct a crime scene [140,146,180,181]. The fertilizer particles require suppression of the coffee-ring effect to enhance uptake, and contamination removal from evaporating droplets is easier in sessile mode than in pendant configuration [140]. The difference in the pattern formed after drying of a colloidal droplet in a sessile, pendant, and inclined mode can be used to make selective electrically conductive carbon networks and contacts for self-assembled nanostructures arrays on solid and flexible polymer substrates [141].

## Figures and Tables

**Figure 1 nanomaterials-12-02600-f001:**
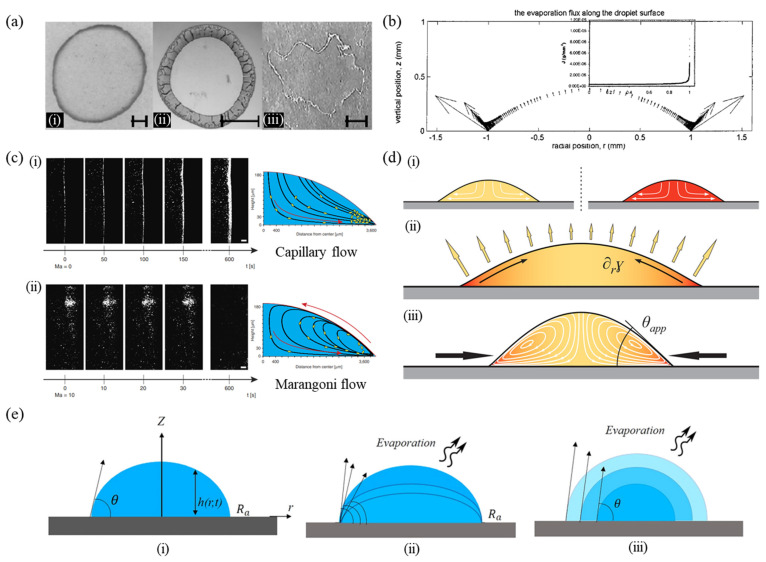
(**a**) Dried patterns of (i) coffee stain, (ii) colloidal microspheres, and (iii) salt deposit. The scale bar is 1 cm (reproduced with permission from ref. [30]. Copyright 2000 American Physical Society). (**b**) Evaporation flux along the droplet surface. The inset represents the magnitude of the evaporation flux along the droplet surface (reproduced with permission from ref. [31]. Copyright 2002 American Chemical Society). (**c**) (i) Capillary flow resulting in a narrow band of bacteria, which accumulates at the edge of the droplet, known as the coffee-ring effect. (ii) Marangoni flow opposing the coffee-ring effect thus creating vortices or swirling of the bacteria in the droplet (reproduced with permission from ref. [112]. Copyright 2013 Springer Nature). (**d**) (i) Sessile droplets of two separate liquid components, one is water (high surface tension), and the other is ethanol (low surface tension). (ii) An evaporating sessile droplet with a binary mixture of the components with different surface tension and volatilities. (iii) The surface tension gradient induced by evaporation drives a Marangoni flow (reproduced with permission from ref. [114]. Copyright 2017 American Chemical Society). (**e**) (i) A sessile droplet on a solid substrate. (ii) CCR mode, and (iii) CCA mode of droplet evaporation on a solid substrate (reproduced with permission from ref. [115]. Copyright 2021 AIP Publishing).

**Figure 2 nanomaterials-12-02600-f002:**
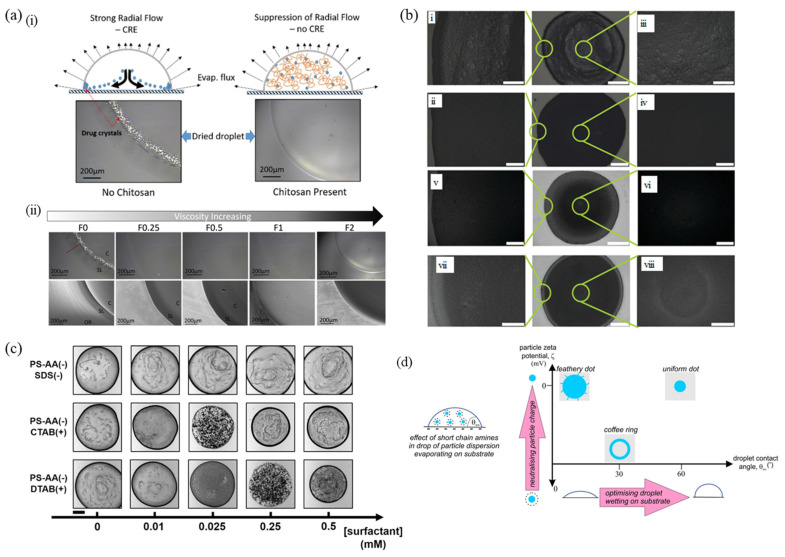
(**a**) (i) Drying process and final pattern of an aqueous droplet of paracetamol with/without chitosan polymer and (ii) Polarising light microscopy/PLM (top) and SEM (bottom) images of the final deposition patterns at the center (C), stagnation line (SL) and outer ring (OR) regions at different chitosan-based formulations, 0, 0.25, 0.5, 1 and 2% (*w*/*w*) respectively (reproduced with permission from ref. [121]. Copyright 2021 AIP Publishing). (**b**) Magnification optical microscope images of SiO_2_ microspheres at droplet edge (i) without PEO1 and (ii) with PEO1 additives; at droplet center (iii) without PEO1 and (iv) with PEO1 additives; with PEO2 additives (v) at droplet edge and (vi) center; and with PVA additives (vii) at droplet edge and (viii) center. The scale bar is 250 μm for (i–viii), and 1 mm for the middle optical microscope images (reproduced with permission from ref. [55]. Copyright 2012 American Chemical Society). (**c**) Microscopic images of deposits formed from evaporating droplets (0.8 μL) of mixtures of anionic PS particles (PS-AA, 500 nm diameter, 2 mg/mL) with surfactants (SDS, CTAB, and DTAB) at various concentrations (scale bar is 500 μm) (reproduced with permission from ref. [43] Copyright 2015 American Chemical Society). (**d**) The effect of short-chain amine adsorption on the silica nanoparticles dispersed in a drop evaporating on a substrate (reproduced with permission from ref. [128]. Copyright 2020 Elsevier).

**Figure 3 nanomaterials-12-02600-f003:**
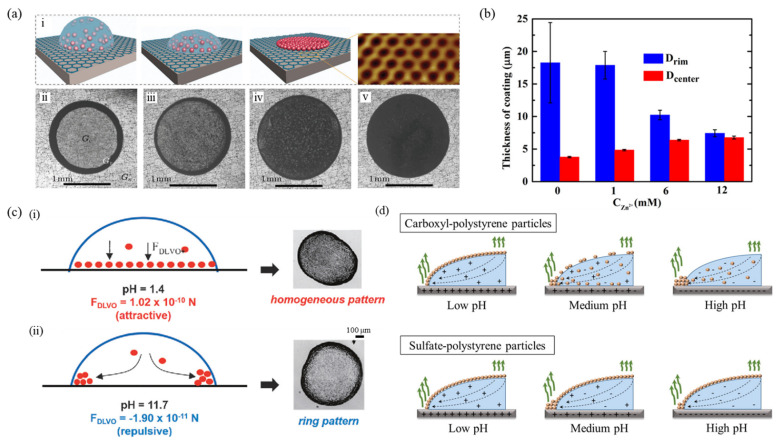
(**a**) (i) Schematic of how cations in a drop (blue hemisphere) determine the deposition of PS microspheres (red spheres). Inset shows an atomically resolved scanning tunneling microscope (STM) image of a graphene lattice. (ii)–(v) Optical microscopy images of particle patterns on graphene after evaporation of mixture drops with different salt concentrations (0 mM, 2.0 mM, 4.0 mM, and 8.0 mM, respectively) (reproduced with permission from ref. [67]. Copyright 2020 Chinese Physical Society and IOP Publishing Ltd.). (**b**) Average thickness at rim and center of PS microsphere suspension mixed with different concentrations of complexed Zn^2+^ (0, 1.0, 6.0, and 12.0 mM), after evaporation of the sessile drops on the iron surface (reproduced with permission from ref. [71]. Copyright 2020 American Chemical Society). (**c**) Influence of pH-dependent DLVO interactions of particles with the liquid-solid interface on the deposit morphology from drops of titania nanoparticles drying on glass substrates at (i) acidic and (ii) basic conditions (scale bar is 100 μm) (reproduced with permission from ref. [117]. Copyright 2010 American Chemical Society). (**d**) Schematic of carboxyl- and sulfate-PS particle deposits at different pH levels (reproduced with permission from ref. [118]. Copyright 2018 Elsevier).

**Figure 4 nanomaterials-12-02600-f004:**
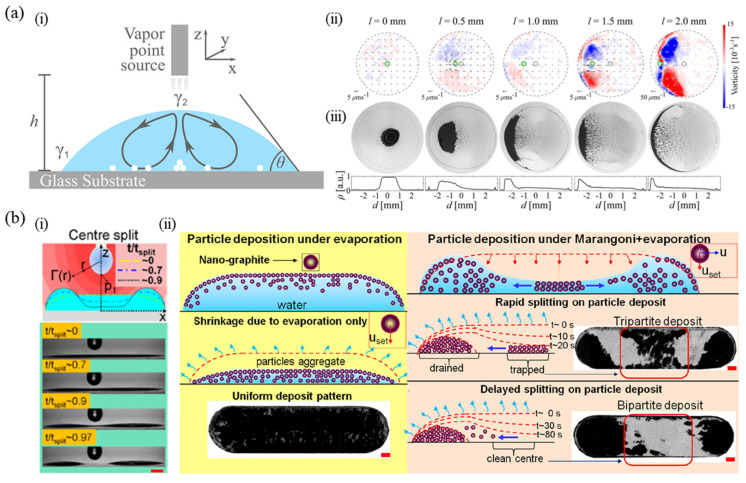
(**a**) (i) Schematic representation of an evaporating sessile droplet under a vapor point source. (ii) Velocity vectors and vorticity by the Marangoni flows as a function of the lateral displacement 𝑙 of the vapor point source from the droplet’s center for *h* = 2 mm. (iii) Final deposits after evaporation of sessile drops. The bottom histograms show the corresponding deposit’s density profile along the droplet’s diameter and the densities were averaged along the angular coordinate (reproduced with permission from ref. [102]. Copyright 2018 American Chemical Society). (**b**) (i) Center split of a long sessile drop by an evaporating ethanol droplet. The scale bar equals 1 mm. (ii) Particle deposition under only evaporation and the combined Marangoni flow and evaporation showing Moses’s effect. Scale bars are 1 mm (reproduced with permission from ref. [103]. Copyright 2020 American Chemical Society).

**Figure 5 nanomaterials-12-02600-f005:**
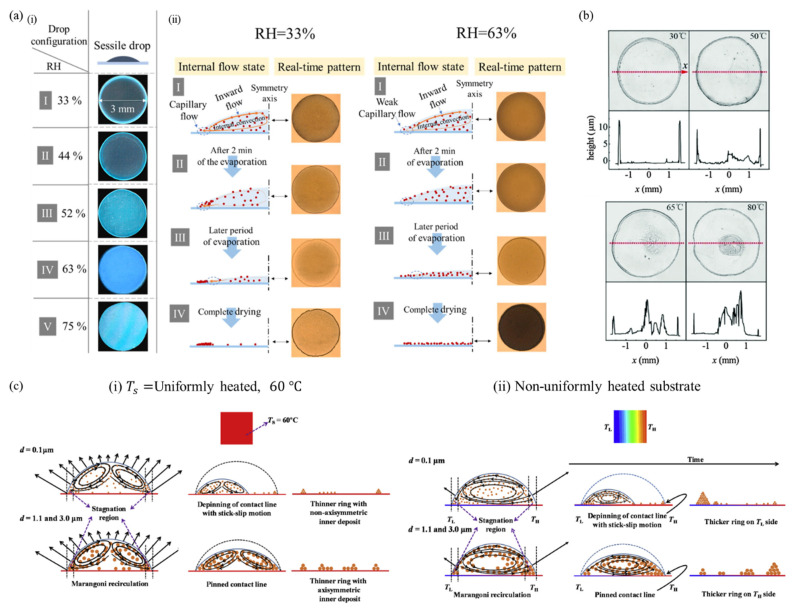
(**a**) (i) Dried patterns of sessile drops at various RH levels of 33%, 44%, 52%, 63% and 75%. (ii) Schematic migration processes of microspheres and real-time patterns during evaporation of the sessile drop under 33% and 63% (reproduced with permission from ref. [73]. Copyright 2022 Elsevier). (**b**) Final deposit pattern of water droplet containing PS particles of 100 nm as a function of the substrate temperature (reproduced with permission from ref. [78]. Copyright 2015 Royal Society of Chemistry). (**c**) Mechanism of the deposit formation on (i) a uniformly heated substrate and (ii) a non-uniformly heated substrate depending on the diameter of particles (reproduced with permission from ref. [79]. Copyright 2020 Elsevier).

**Figure 6 nanomaterials-12-02600-f006:**
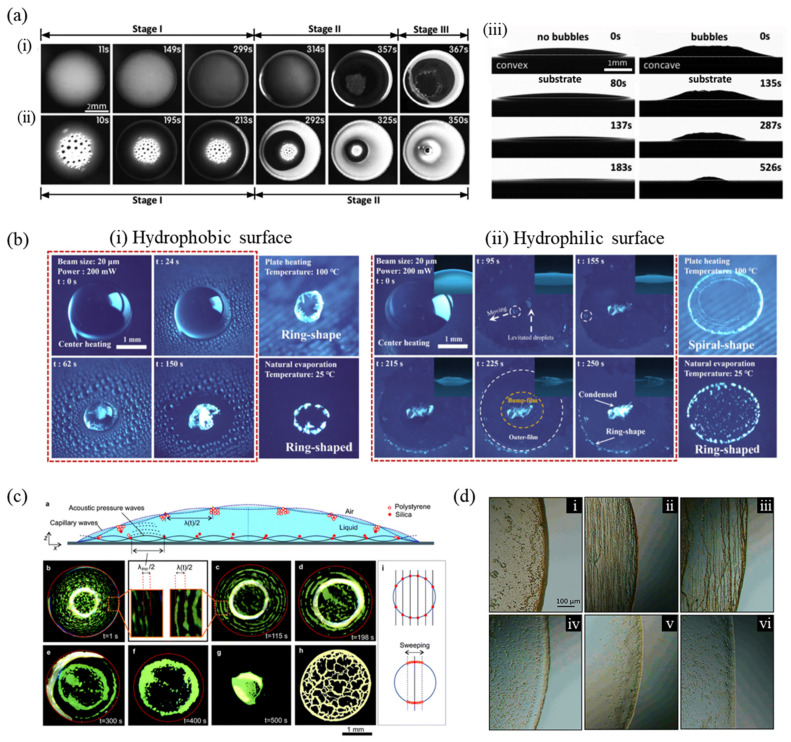
(**a**) Drying procedure of droplets containing polytetrafluoroethylene (PTFE) particles of 60 wt.% (i) without and (ii) with air bubbles on monocrystalline silicon wafers. (iii) Side views of the drying process of a 5  μL droplet with and without bubbles (reproduced with permission from ref. [135]. Copyright 2020 Elsevier). (**b**) Evaporative crystallization process of the light-induced droplet and final deposition patterns under the light-induced, plate-heating-induced, and natural evaporation on the (i) hydrophobic and (ii) hydrophilic surfaces (reproduced with permission from ref. [136]. Copyright 2021 American Chemical Society). (**c**) Schematic illustration of a droplet excited by SAW and images showing the drying patterns of a colloidal drop under SAW with time (reproduced with permission from ref. [52]. Copyright 2015 Royal Society of Chemistry). (**d**) Effect of the applied magnetic field on the ring thickness. (i) 0 T, (ii) 0.005 T, (iii) 0.02 T, (iv) 0.07 T, (v) 0.1 T, and (vi) 0.15 T (reproduced with permission from ref. [107]. Copyright 2017 AIP Publishing).

**Figure 7 nanomaterials-12-02600-f007:**
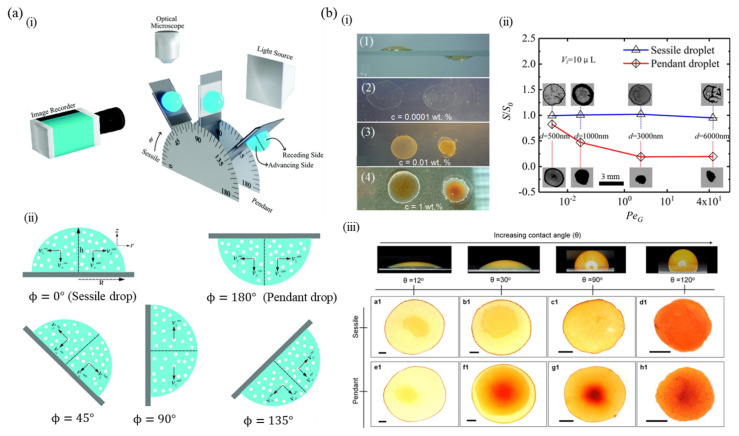
(**a**) (i) Schematic of the experimental set-up used to generate colloidal dispersion drops on substrates placed at different inclinations. CCD camera (image recorder) captures a side view and top view of the drop and an optical microscope is used to study the kinetics of drop evaporation, and (ii) illustrations show the components of velocity experienced by particles in a drop drying at different substrate inclinations: the direction of particle velocity parallel and perpendicular to the substrate in advancing side (respectively labeled as $v_{∥}^{adv}$ and $v_{⊥}^{adv}$) and receding side (respectively labeled as $v_{∥}^{rec}$ and $v_{⊥}^{rec}$) (reproduced with permission from ref. [147]. Copyright 2021 Royal Society of Chemistry). (**b**) Comparison of patterns obtained from the drying drops of colloidal dispersions in sessile and pendant modes for (i) (1) 22 nm iron oxide particles (initial contact diameter ∼8 mm) deposited, and (2), (3) and (4) dried on a glass substrate at different concentrations of 0.0001 wt.%, 0.01 wt.% and 1 wt.%, respectively (reproduced with permission from ref. [141]. Copyright 2011 Elsevier). (ii) Drops (10 μL) containing PS colloidal particles (0.1 wt.% and initial contact radius of 1.5 mm) of different particle sizes. The scale bar is 3 mm (reproduced with permission from ref. [149]. Copyright 2018 American Chemical Society). (iii) Drops (2 μL) containing hematite ellipsoids (∼59 nm diameter, ∼244 nm long) at a concentration of 0.12 wt.% on substrates of different wettabilities, namely θ = 12 ± 3°, 30 ± 5°, 90 ± 2°, and 120 ± 5°. The scale bar is 500 μm (reproduced with permission from ref. [143]. Copyright 2018 American Chemical Society).

**Figure 8 nanomaterials-12-02600-f008:**
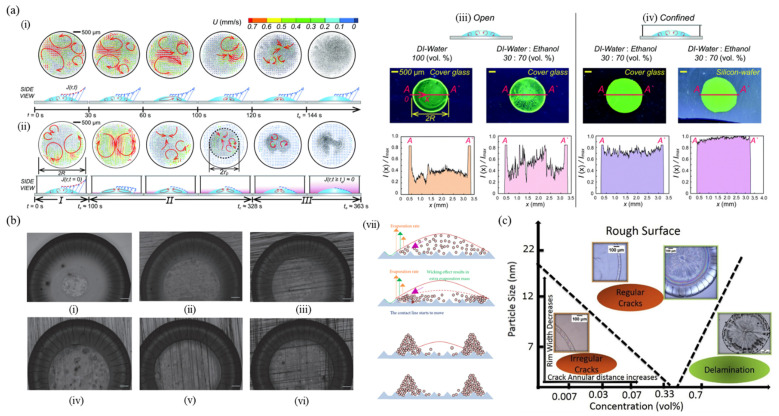
(**a**) Internal flow fields of an ethanol–water (70:30 vol%) mixture droplet (i) on an open and (ii) in confined geometry. Deposition patterns of quantum dots (QDs) (iii) on an open and (iv) in a confined space (reproduced with permission from ref. [94]. Copyright 2021 Royal Society of Chemistry). (**b**) (i–vi) crack formation in the deposit patterns of PTFE colloidal droplets on glass slides with different roughness: (i) smooth glass and (ii–vi) increasing surface roughness which is obtained using sandpaper with different grit sizes (1500, 1000, 800, 400, and 200, respectively). Scale bars represent 100 μm. (vii) Evaporation mechanism of a colloidal droplet on a rough substrate (reproduced with permission from ref. [99]. Copyright 2018 Hindawi). (**c**) Phase diagram for a nano rough surface considering the effects of particle size, concentration, and substrate morphology (reproduced with permission from ref. [100]. Copyright 2020 Elsevier).

**Figure 9 nanomaterials-12-02600-f009:**
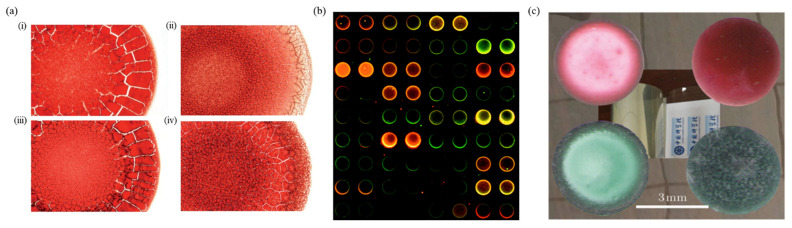
(**a**) Dried drops of blood (drop diameter: 5.9 mm) in (i) sample from a 27-year-old woman in good health, (ii) person with anemia, (iii) sample from a 31-year-old man in good health and (iv) person with hyperlipidemia (reproduced with permission from ref. [155]. Copyright 2010 Cambridge University Press). (**b**) Non-uniform deposition patterns in cDNA microarray (reproduced with permission from ref. [166]. Copyright 2002 American Chemical Society). (**c**) Photographs of patterns on a PET film after evaporation of drops of mixtures of acid red 1 (upper) and acid blue 25 (lower) solutions with 0 mM (left) and 16.0 mM (right) NaCl, respectively (reproduced with permission from ref. [67]. Copyright 2020 Chinese Physical Society and IOP Publishing Ltd.).

## Data Availability

Not applicable.

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
