# Peer review of "Control of the Drying Patterns for Complex Colloidal Solutions and Their Applications"

_nanomaterials, 2022, doi:10.3390/nano12152600_

Round 1
Reviewer 1 Report
In this manuscript, the recent advances of alleviation the coffee-ring effect of dry colloidal solution droplet were reviewed from the aspects of adding chemical additives, applying external sources, and manipulating geometrical configurations etc. The content of manuscript is comprehensive and is logically written.
The comments for this work are as follows:
1) Page 5, line 176, the title of “chemical methods” is inappropriate. Because the addition of additives in colloidal solutions does not involve a chemical reaction.
2) It is required more examples or explanations for about the features of colloidal solutions.
3) The challenges and research directions of this research field should be required.
4) Page 20 line 773, It would be better if the summary and the application parts were written separately.

Reviewer 2 Report
This manuscript reviews the complex colloidal solutions related coffee ring phenomenon and applications accordingly. This work is interesting but the authors need to answer the questions below and reinforce their difference with other review work before publishing.
(1) The author mentions during past few decades there are extensively study. So what is this work's new study and contribution? Authors are suggested to discuss somehow the control in the abstract. The abstract now looks the same as other work.
(2) For the final chapter 6, why does authors combine 'summary' and 'applications' together? The title has 'their applications' but authors only discussed in the final chapter. Need an explanation. Also, is figure 9 really required in this summary chapter? It only discussed one sentence for this figure.
(3) For the final chapter 6, the summary discussed a lot of applications. But I feel it is kind of lacking 'summary'. If the authors would like to put summary and applications in one chapter, I suggest to discuss relation between applications and previous chapter 2-5 study and have a final summary paragraph. Now the summary part is not that clear, not like what page 70-76 described.
(4) Some subtitles for chapter 3 could be adjusted. one word 'polymers' , 'surfactants', 'salt' ,'pH' is not clear enough for explaining chemical methods.
Round 2
Reviewer 2 Report
This review work has been revised carefully and answered all the questions. Herein I suggested to publish.